# Wearable Devices and Digital Biomarkers for Optimizing Training Tolerances and Athlete Performance: A Case Study of a National Collegiate Athletic Association Division III Soccer Team over a One-Year Period

**DOI:** 10.3390/s24051463

**Published:** 2024-02-23

**Authors:** Dhruv R. Seshadri, Helina D. VanBibber, Maia P. Sethi, Ethan R. Harlow, James E. Voos

**Affiliations:** 1Department of Bioengineering, Lehigh University, Bethlehem, PA 18015, USA; 2Department of Biomedical Engineering, Case Western Reserve University, Cleveland, OH 44106, USA; 3Sports Medicine Institute, University Hospitals Cleveland Medical Center, Cleveland, OH 44106, USA; hdv7@case.edu (H.D.V.); ethan.harlow@uhhospitals.org (E.R.H.); james.voos@uhhospitals.org (J.E.V.); 4Department of Orthopaedic Surgery, University Hospitals Cleveland Medical Center, Cleveland, OH 44106, USA; 5Department of Cognitive Science, Case Western Reserve University, Cleveland, OH 44106, USA; mxs1496@case.edu; 6Department of Psychology, Case Western Reserve University, Cleveland, OH 44106, USA

**Keywords:** wearable technology, musculoskeletal modeling, digital biomarkers, human performance, biomedical engineering

## Abstract

Wearable devices in sports have been used at the professional and higher collegiate levels, but not much research has been conducted at lower collegiate division levels. The objective of this retrospective study was to gather big data using the Catapult wearable technology, develop an algorithm for musculoskeletal modeling, and longitudinally determine the workloads of male college soccer (football) athletes at the Division III (DIII) level over the course of a 12-week season. The results showed that over the course of a season, (1) the average match workload (432 ± 47.7) was 1.5× greater than the average training workload (252.9 ± 23.3) for all positions, (2) the forward position showed the lowest workloads throughout the season, and (3) the highest mean workload was in week 8 (370.1 ± 177.2), while the lowest was in week 4 (219.1 ± 26.4). These results provide the impetus to enable the interoperability of data gathered from wearable devices into data management systems for optimizing performance and health.

## 1. Introduction

In men’s professional soccer, there is an injury incidence of 8.7 per 1000 h of exposure, including a high percentage of hamstring (12.3%), ankle (8.5%), and adductor (7.6%) strains [1]. Soft tissue injuries in athletes can result in either time loss or no time loss, meaning that the athletes are able to play through their injuries without missing training or competitive competitions [2]. Male professional athletes missed a mean time of 15.8 days due to injury, while 44.2% of injuries did not result in days missed over the course of a 6-year prospective study [1]. In collegiate sports, about half of the injuries in men’s and women’s sports are non-time-loss injuries [2]. Furthermore, at the collegiate level, men have higher acute injury rates (49.8 vs. 38.6 for every 10,000 athlete exposures) but lower rates of overuse injury (24.6 vs. 13.2 for every 10,000 athlete exposures) compared to women [3]. It is hypothesized that, compared to male athletes, female athletes are at a greater risk for overuse injuries (e.g., general stress, inflammation, tendinitis) since men have larger patellar tendons after consistent training, higher rates of collagen synthesis, and stronger collagen fascicles [3,4,5].

At the collegiate level in the United States, universities are divided into divisions: Division 1 (DI), Division 2 (DII), and Division 3 (DIII). National Collegiate Athletic Association (NCAA) DI schools receive about 60% of the NCAA’s annual revenue [6]. DII and DIII schools receive 4.37% and 3.18% of the revenue, respectively [6]. The staffing at each of these divisions, moreover, reflects this inequality. DI schools have larger staffing capabilities (part-time and full-time athletic trainers) than all other competition levels do [7,8]. DI athletics have been indicated to have increased numbers of athletic training, satellite athletic training, physical therapists, and game-day facilities [8,9]. In terms of injury rates, DII and DIII schools were found to have significantly higher rates of bone stress injuries (BSIs) than those of DI athletes (1.21; 95% CI, 1.01–1.44), as well as a significant difference in time lost due to injury (χ^2^ = 16.54; *p* = 0.006) [10]. It is hypothesized these health and injury disparities among collegiate athletes are a result of a lack of access to facilities and training equipment at lower-division schools.

Athletic programs at both the professional and amateur levels have shifted toward using wearable sensors as a primary means of collecting objective, continuous data to quantify training load with the intention of assessing health performance and reducing injury burden [11,12,13,14,15,16,17]. The use of wearable sensors is commonly seen at the professional and DI level with the use of wearable GPS sensors to quantify training loads and monitor injuries throughout several seasons [18,19,20,21,22,23,24]. The acute-to-chronic-workload ratio (ACWR) is considered the “gold” standard for modeling training loads and injuries [25]. The player load can be calculated through an analytical framework that measures acceleration in the *x*, *y*, and *z* directions using a tri-axial accelerometer (Equation (1)) [12].
(1)$$\;\mathrm{Player}\;\mathrm{Load}=\frac{\sqrt{{\left(a_{x\left(t\right)}-a_{x\left(t-1\right)}\right)}^{2}+{\left(a_{y\left(t\right)}-a_{y\left(t-1\right)}\right)}^{2}+{\left(a_{z\left(t\right)}-a_{z\left(t-1\right)}\right)}^{2}}}{100},$$

Data for the ACWR are collected via wearable technology in sports and are contrived by dividing the acute workload (fatigue) by the chronic workload (fitness) [25]. The acute workload is the workload of the desired week, such as the week preceding an injury [25]. The chronic workload is the average of the workloads for the four-week period preceding the acute workload week [25]. The coupled ACWR includes the acute workload in the chronic workload calculation, while the uncoupled ACWR does not [12]. Li et al. studied the relationship between external workload, which was measured using the Catapult OptimEye S5 (Catapult Innovations, Melbourne, Australia), and soft-tissue injury in professional football athletes over a two-year period [18]. Athletes who had an ACWR greater than 1.6 were 1.5 times more likely to suffer either a myotendinous or ligamentous injury [18]. This was the first published study in professional football to show this correlation, suggesting that players should be monitored weekly to screen for athletes who may be approaching vulnerable ACWR levels [18]. While the ACWR is widely accepted, Impellizzeri et al. suggested that the ACWR and its components be avoided [25]. Members of this team determined that the ACWR has no predictive advantage in terms of injury, magnifies the acute workload, has a lower standard deviation that leads to higher odds ratios, and does not predict injury [25]. GPS sensors have additionally been used at the DI level of collegiate soccer athletes to adapt athletes to the training and game workload expectations of a standard soccer season and reduce injury risk. Bertschy et al. utilized GPS technology (Titan 1) with the goal of creating an acclimation program for summer workouts to reduce the rate of incidence of musculoskeletal injuries [26]. The team monitored 27 (year 1) and 29 (year 2) DI female collegiate soccer athletes over the course of two seasons [26]. The data gathered from year 1 were utilized to create a 5-week GPS summer acclimation program for year 2 [26]. The results of the study indicated that compared to year 1, the inclusion of the GPS-guided acclimation program in the summer before year 2 resulted in a 50% reduction in injury prevalence [26].

Research on the use of wearable sensors has been present at the lower NCAA collegiate levels (DII and DIII); however, there is a lack of longitudinal data from wearable technology used to monitor workload profiles and correlate them with potential injury risk [27,28]. In one study, twenty-two DIII female soccer athletes wore Polar GPS sensors over the course of a season, which included 47 practices and 22 matches [27]. The results showed that starters had almost double the training load and a higher total distance traveled compared to reserves [27]. Extremely high training loads increase the risk of overtraining, fatigue, and injury [27]. Extremely low training loads can also lead to problems, such as insufficient fitness and a higher risk of injury [27]. Analyzing and managing player workloads may address imbalances between starters and reserves [27]. While researchers speculate that the management of training loads can potentially reduce the risk of injury, further assessment of injury risk and prevalence is lacking in lower collegiate divisions [12,21]. 

The highlighted gap in current orthopedic sports performance research points to the financial inequality across collegiate athletics [6,7,8,29]. The financial inequality among the DI, DII, and DIII collegiate athlete divisions, therefore, translates into the importance and need for the engineering and development of low-cost sensor technology and open-access algorithms that are amenable to teams in lower collegiate levels to enable more equitable sports medicine monitoring. Table 1 provides comparative analysis of the cost of current wearable GPS sensors used in sports today. Because of the limited funding and resources for DIII athletics, the authors herein stress the need for collaboration among coaches, sports scientists, team physicians, and athletic trainers to maximize performance and reduce injury burden [6]. While research has been performed to explore the relationship between workloads and injuries and create an algorithm for injury prevention at the professional level and DI level, there is an increasing demand for attention at the lower collegiate division levels [18,26]. Thus, there remains an unmet medical need to study and monitor athletes at lower divisions to gain a better understanding of what perpetuates these differences to ultimately optimize performance and mitigate injuries. The primary objective of this study was to gather normative baseline data on healthy male collegiate athletes at the DIII level toward the development of future performance optimization protocols.

## 2. Materials and Methods

Data were provided by the Case Western Reserve University (CWRU) soccer team for analysis and were subsequently de-identified prior to analysis. This longitudinal retrospective study occurred over the course of the 2019 fall season for the DIII collegiate men’s soccer team. A total of 28 athletes participated over the 12-week season. Each participant was classified into position groups: 2 attacking midfielders (AMFs), 4 center backs (CBs), 4 forwards (Fs), 4 goalkeepers (GKs), 4 left backs (LBs), 4 midfielders (MFs), 2 right backs (RBs), and 4 wingers (Ws). The participants in this study wore the same Catapult PlayerTEK for each training session and competitive match. The players voluntarily participated with the understanding that they could remove the device at any point. The sensor was located between the player’s scapulae and was secured a harness. After each training session, the sensors were turned in, and the data were downloaded via the corresponding software platform.

Player workloads were determined throughout the course of the season during practice and matches. The mean player workload was determined over each training and match session and over each week throughout the season. The coupled versus uncoupled ACWR was calculated using the Rolling Average method. Data were analyzed using Microsoft Excel Version 16.16.27. Normalization was performed by generating T-values to normalize the workload profiles to the mean workload.

## 3. Results and Discussion

Twenty-eight participants (BMI 22.5 ± 1.2 kg/m^2^) completed the study over the 12-week season. Edge computing from the Catapult PlayerTEK Workload profiles was assessed across the team. Movement profiles were obtained through accelerometer and GPS receiver movements. The mean workload for each training and match session for the overall team during the season is displayed (Figure 1). Throughout the 59 sessions assessed in the season, session 23 illustrated the lowest mean workload (171.6 ± 17.8) (Figure 1a). The highest mean workload was during session 38 (603.2 ± 150.8), displaying a 3.5× workload difference. The mean workload was the lowest in week 4 (219.1 ± 26.4), while the highest mean workload was in week 8 (370.1 ± 177.2) (Figure 1b). A workload difference of 1.7× was displayed between week 4 and week 8. At week 9, the mean workload decreased (319.6 ± 106.7) from the highest workload seen in week 8 (Figure 1b). A qualitative color gradient using green for low workloads (10th percentile), yellow for medium workloads (50th percentile), and red for high workloads (90th percentile) permits the efficient viewing of the differences in the weekly workload during the study (Figure 1c). As illustrated by the qualitative plot, the workload was low between weeks 2 and 7, but by week 12, the mean workload for the team progressed to a high workload (Figure 1c).

The workload profiles were assessed positionally (Figure 2). Week 12 displayed the highest mean workload for the following positions: AMF (483.8 ± 67.7), CB (355.5 ± 127.3), F (339.4 ± 73.8) LB (411.2 ± 159.2), MF (442.7 ± 82.3), and W (443.9 ± 82.3) (Figure 2b). The RB position had the highest mean workload at week 8 (488.1 ± 274.1). The workload difference between the highest mean workload in this week (AMF) and the lowest mean workload was 1.4×. The cumulative workloads for each position across the duration of the season are illustrated (Figure 2c). The forward position (F) displayed the lowest cumulative workload compared to the other positions. The positional workload profiles between practice and match sessions in the team were recorded (Figure 3). The average match workload (432.9 ± 47.7) was much greater than the average training workload (252.9 ± 23.3) for the entire team, displaying a workload difference of 1.8× (Figure 3a). Positionally, the forward position (F) delineated the lowest workload for both matches (351.1 ± 113.4) and practice (237.1 ± 43.8). The match workload was notably greater than the training workload for all positions (Figure 3b). The match-to-practice workload was measured for each position: AMF (1.5), CB (1.8), F (1.5), LB (1.8), MF (1.8), RB (1.9), and W (1.6). The difference between the highest match-to-practice workload (RB) and the lowest match-to-practice workload (F) was 1.3×.

Figure 4 depicts the normalized workload profiles for each position group (AMF, CB, L, LB, MF, RB, and W) for each week of the season. T-values were generated to compare the normalized workloads to the team’s mean workload (306.11 ± 55.6) (Figure 4a). The null hypothesis was 0, indicating that there was no difference between the team’s mean workload and the workload for the respective position group’s workload. A greater positive magnitude suggests a higher average workload compared to the mean, whereas a greater magnitude of the negative t-values indicates a negative deviation from the mean workload and, thus, a lower workload. A qualitative plot is used to illustrate the data from Figure 4a (Figure 4b). Week 4 illustrated the most negatively deviated t-values with respect to the mean workload for the positions AMF, CB, LB, MF, and W, which displayed respective t-values of −7.1, −8.2, −8.96, −6.4, −8.4, and −6.9. Week 12 displayed the lowest t-value for the RB position (−7.5). Negative workloads were most strongly represented in Week 4, as shown by the red color (Figure 4b). Week 12 showed the most positive deviations from the team’s mean workload for the following positions: AMF (+15.7), CB (+4.4), F (+2.9), LB (+9.3), MF (+12.0), and W (+12.1). The magnitude of the positive t-values in week 12 indicated a positive deviation from the team’s mean workload and, thus, a higher workload. The blue color in Week 12 illustrates this trend (Figure 4b). Week 8 showed the highest t-value for the RB position (+16.0). The coupled and uncoupled ACWRs across a 3-week chronic load were determined using the Rolling Average method (Figure 5). A color-gradient platform correlated with the ACWR is used to illustrate each position’s coupled ACWR from week 3 to week 12 (Figure 5a). Low workloads (green) were seen in weeks 3, 4, 9, and 12 (Figure 5a). In the athletes, during these weeks, the ACWR was <1.0. A medium workload (yellow) was seen when the ACWR approached 1.0. A high workload (red), however, was prominent during weeks 5, 6, and 10 (Figure 5a). Here, a high workload indicated that the athletes’ ACWR exceeded 1.0, and some athletes approached or exceeded an ACWR of 1.5. A qualitative plot representing the data from Figure 5a was created to assess the normality of the workload data across each position group throughout the season (Figure 5b). The red color for week 12 presented the most negative deviations from the mean workload and, thus, lower workloads. As shown by the color gradient, week 6 portrayed significantly high workloads, as represented in the normalized plot, with the blue color indicating the increased positive deviations from the mean ACWR and, therefore, increased ACWRs in this week (Figure 5b).

The uncoupled ACWR with the chronic workload was calculated using the Rolling Average method across a 3-week period beginning at week 3. Here, a low ACWR (<1.0) was seen in weeks 3, 4, 9, and 12. A high ACWR (>1.0) was evident in weeks 5, 6, and 10 (Figure 6a). A qualitative plot is used to illustrate the normalization of the ACWR during weeks 3 and 12 from the data in Figure 6a (Figure 6b). Week 6 illustrated the most positive deviations from the mean workload for all positions, but this was most overtly portrayed for the CB position. Similarly to the findings for the coupled ACWR, week 12 presented the most negative deviations from the mean ACWR and, thus, the lowest ACWRs for all positions (Figure 6b). The differences in the ACWR for the coupled and uncoupled calculations are visualized for each position group across weeks 3 to 12 (Figure 7).

The coupled ACWR was calculated when the chronic workload was assessed over a 4-week period using the Rolling Average method. The correlation of the ACWR for each position group during each week is visualized on a color-gradient platform (Figure 8a). For all position groups, a lower ACWR was seen in weeks 4, 9, and 12. The ACWR for these weeks reflected lower workloads, which ranged from 0.23 to 1.04. High ACWRs were seen in weeks 5 and 6 for all position groups. The qualitative plot portraying the normalized workload data further indicates that each position group experienced a high workload relative to the mean workload. The normalization of the data from Figure 8a indicated that week 12 illustrated the most negative deviations from the mean workload, as illustrated by the red color for each position group (Figure 8b). On the contrary, both week 5 and week 6 showed positive deviations from the mean ACWR, as visualized by the blue color for each position group (Figure 8b).

The uncoupled ACWR was calculated when the chronic workload was calculated over a 4-week period. The correlation of the ACWR for each position is visualized on a color-gradient platform (Figure 9a). For all position groups, low workloads were seen in weeks 4, 9, and 12. The ACWR during these weeks ranged from 0.19 to 1.03 (excluding the LB position in week 9). Week 6 showed similar and high uncoupled ACWRs for all positions. The normalization of the workload data is qualitatively visualized (Figure 9b). Week 12 illustrated the most negative deviations from the mean ACWR, as visualized by the red color in the normalization plot. Week 6 portrayed positive deviations from the mean ACWR, as illustrated by the blue color for all positions (Figure 9b). The differences between the uncoupled and coupled ACWRs across the 4-week period for each position and week were compared (Figure 10). For the AMF, CB, F, MF, and W positions, both the coupled and uncoupled ACWR were highest at week 6. The uncoupled ACWR exceeded the coupled ACWR here—for example, in the AMF position, the uncoupled workload at week 6 was 1.68 and the coupled ACWR was 1.43.

The relationship between the coupled and uncoupled workloads when the chronic workload was measured over a 3-week period was determined (R^2^ = 0.92, *p* = 0.10) (Figure 11a). The relationship between the uncoupled ACWR and coupled ACWR across a 4-week period was found (R^2^ = 0.78, *p* = 0.17) (Figure 11b). A weak relationship between the coupled ACWR across the 3-week chronic workload and the coupled ACWR across the 4-week chronic workload was additionally determined (R^2^ = 0.04, *p* = 0.41) (Figure 11c). The relationship between the uncoupled ACWR across the 3-week chronic workload and the uncoupled ACWR across the 4-week chronic workload was examined (R^2^ = 0.05, *p* = 0.39) (Figure 11d).

One athlete was selected by utilizing a randomization algorithm. A randomized array computed via the Rolling Average model showed the mean workload of the athlete compared to the mean workload of the respective position; this is visualized by using green for the mean workload of the athlete and yellow for the mean workload in that position in a spider chart (Figure 12a). Similarly to the findings in the qualitative plots, the ACWR for the specific athlete was compared to the mean ACWR for the entire team when the chronic workload was 3 weeks (Figure 12b).

The visualization here illustrates that the ACWRs for the specific athlete and for the entire team were similar, as overlaps are displayed in the spider charts. The comparative profile of the ACWR when the chronic workload was 3 weeks was additionally visualized (Figure 12c).

## 4. Conclusions

The objective of this retrospective study was to ascertain differences in workload profiles between practice and matches and develop injury risk stratification analytics to help guide performance and training in Division III collegiate student athletes. The data for this study were collected using the Catapult PlayerTEK device, which was worn by the student athletes during all training and matches. Injury data were not collected by the team. The study demonstrated that for all positions, matches required increased workloads compared to training sessions (1.8×). This finding was consistent with those in the previous literature, which suggested that matches require increased workloads compared to training sessions [19,30]. Among all seven positions studied, the forward (F) position exhibited the lowest workloads in both matches (1.2× lower than the average match workload) and practice (1.1× lower than the average training workload) throughout the season. The difference in the F position may be attributed to differences in positional requirements, such as distance and time requirements [31,32]. Thus, examination of the workloads and responsibilities of each position could be used to optimize performance assessments for each position; thus, this was a focus of this work.

The relationship between the uncoupled and coupled ACWRs using the 3-week (21 day) chronic workload—a strong, positive relationship—presented findings consistent with the previous exploration of this relationship [12]. The contextualization of the workload with ACWRs for each position has implications for understanding the propensity for injury risk in the athletes in this study. A high workload relative to each position was indicated by a ‘red danger zone’ visual (90th percentile). Thus, as the high ACWR varied positionally, this further suggested how differing positional requirements may translate into differing training and match workloads. The relationship between the ACWR and injury risk in NCAA DIII collegiate athletes has been studied less. The difference between lower collegiate and higher collegiate/professional levels emphasizes the speculation of a relationship between the ACWR and injury risk but points to a current lack of research that further assesses this relationship [12,21]. The determination of the ACWRs throughout the 12-week season in this study of healthy DIII collegiate athletes provides an initial understanding of this relationship. Our study additionally points to how these differences in injury risk assessment could be mitigated to optimize increased performance protocols at this sports level.

## 5. Future Work

The implications of this study further suggest the potential to create an athlete management system for coaches, athletic trainers, team physicians, and athletes to follow in future seasons (Figure 13). This collaborative effort with the implementation of objective workload data, however, requires the development of clinically meaningful algorithms that can translate data from commercial devices, such as the Catapult. The qualitative gradient of red, yellow, and green to depict workloads and normalizations of the workloads, as presented in this study, serves as a means of easily visualizing the musculoskeletal modeling of an individual’s workload. While sports were the platform for this study, the measurement of acceleration and workload can further be translated to patients with neuromuscular diseases where changes in gait behavior are known to affect quality of life.

## Figures and Tables

**Figure 1 sensors-24-01463-f001:**
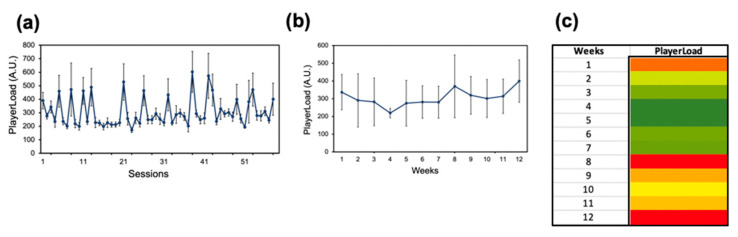
PlayerLoad for the CWRU soccer team. (**a**) Mean workload in each training session and match session. (**b**) Mean workload over each week, including training and matches. (**c**) Heat map generated to correlate workloads into a qualitative color-gradient platform for efficient viewing.

**Figure 2 sensors-24-01463-f002:**
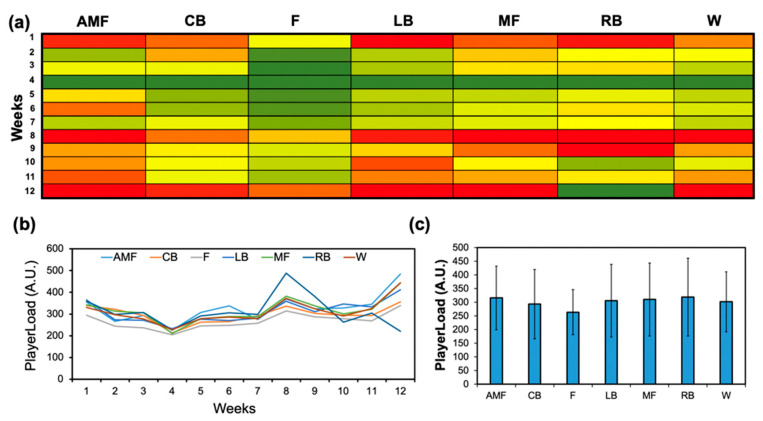
PlayerLoad profiles for each position group on the CWRU soccer team. (**a**) Heat map correlating the mean workloads of each position group over each week into a qualitative color-gradient platform for efficient viewing. (**b**) Mean workload trends for each position group over each week. (**c**) Cumulative workload profiles for each position group considering all 12 weeks of the season.

**Figure 3 sensors-24-01463-f003:**
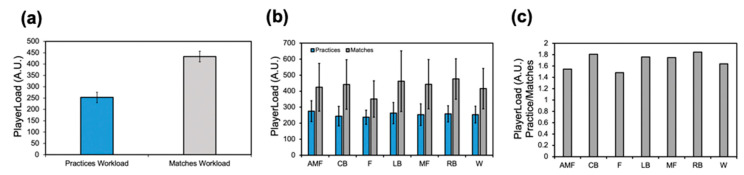
PlayerLoad profiles for each position group on the CWRU soccer team. (**a**) Comparative bar plot assessing workload differences between practice and matches for the entire team. (**b**) Comparative bar plot assessing workload differences between practice and matches for each position group on the entire team. (**c**) The ratio between the match and practice workloads as a function of the various position groups on the team.

**Figure 4 sensors-24-01463-f004:**
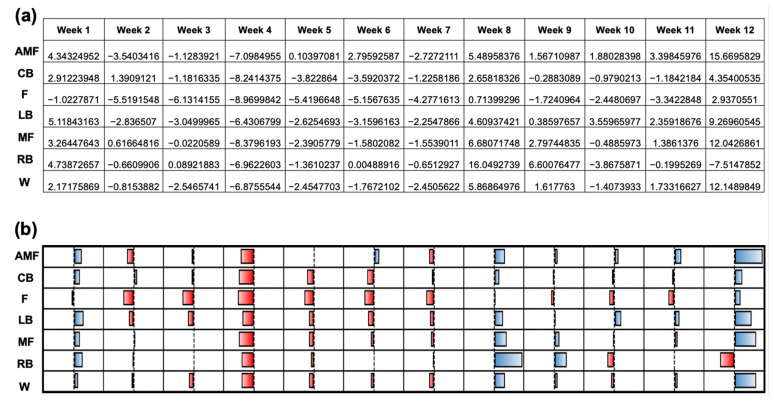
Normalization of the workload profiles per position group for each week of the season. (**a**) T-values were generated to normalize the workload profiles to the mean workload for the entire team over the duration of the season. (**b**) Qualitative plot illustrating data from (**a**) to assess the normality of the workload data among the various position groups over the duration of the 12-week season.

**Figure 5 sensors-24-01463-f005:**
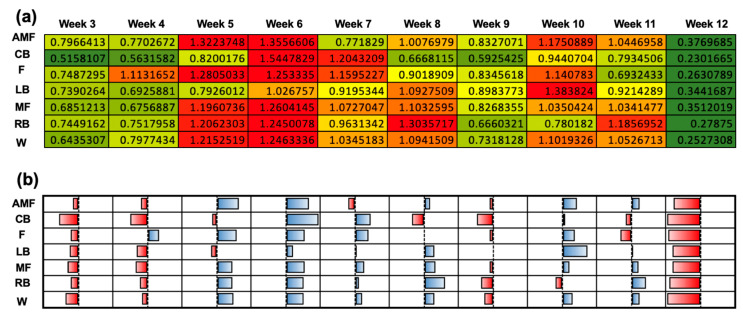
Coupled ACWR using the Rolling Average method when the chronic workload was calculated over a 3-week period. (**a**) Heat map correlating the ACWRs of each position group over each week into a qualitative color-gradient platform for efficient viewing. (**b**) Qualitative plot illustrating data from (**a**) to assess the normality of the workload data among the various position groups over the duration of the 12-week season.

**Figure 6 sensors-24-01463-f006:**
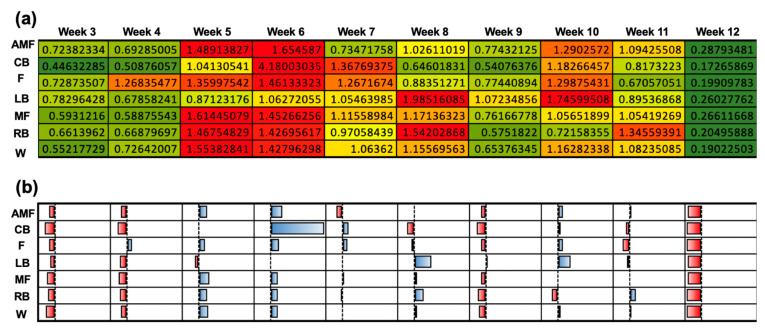
Uncoupled ACWR using the Rolling Average method when the chronic workload was calculated over a 3-week period. (**a**) Heat map correlating the ACWRs of each position group over each week into a qualitative color-gradient platform for efficient viewing. (**b**) Qualitative plot illustrating data from (**a**) to assess the normality of the workload data among the various position groups over the duration of the 12-week season.

**Figure 7 sensors-24-01463-f007:**
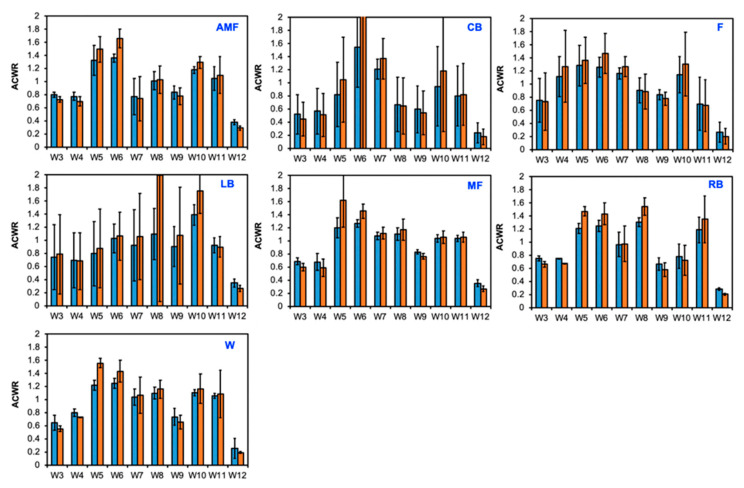
Coupled versus uncoupled ACWR for each position group when the chronic workload was measured over a 3-week span. Blue: coupled; orange: uncoupled.

**Figure 8 sensors-24-01463-f008:**
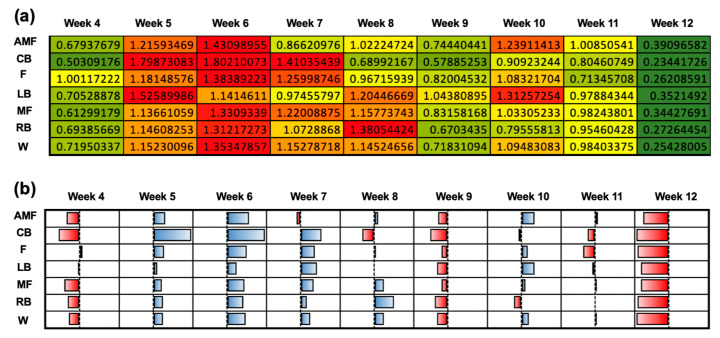
Coupled ACWR using the Rolling Average method when the chronic workload was calculated over a 4-week period. (**a**) Heat map correlating the ACWRs of each position group over each week on a qualitative color-gradient platform for efficient viewing. (**b**) Qualitative plot illustrating data from (**a**) to assess the normality of the workload data among the various position groups over the duration of the 12-week season.

**Figure 9 sensors-24-01463-f009:**
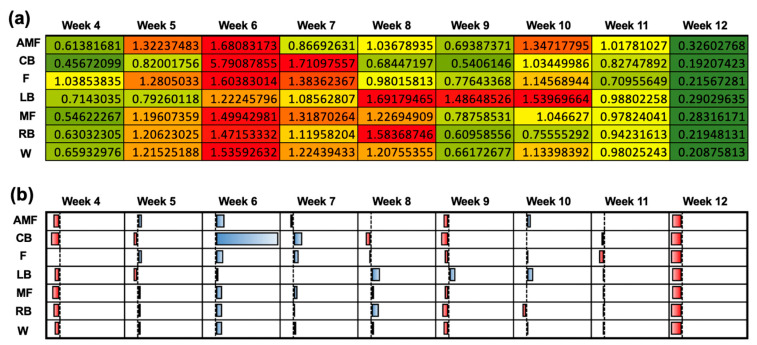
Uncoupled ACWR using the Rolling Average method when the chronic workload was calculated over a 4-week period. (**a**) Heat map correlating the ACWRs of each position group over each week on a qualitative color-gradient platform for efficient viewing. (**b**) Qualitative plot illustrating data from (**a**) to assess the normality of the workload data among the various position groups over the duration of the 12-week season.

**Figure 10 sensors-24-01463-f010:**
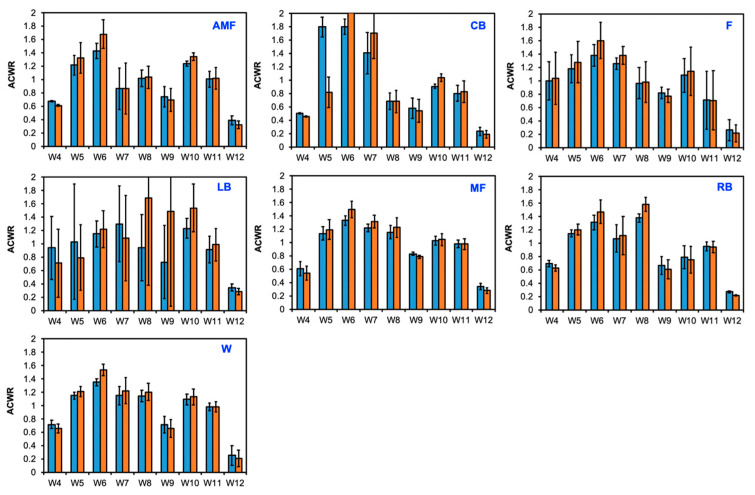
Coupled versus uncoupled ACWR for each position group when the chronic workload was measured over a 4-week span. Blue: coupled; orange: uncoupled.

**Figure 11 sensors-24-01463-f011:**
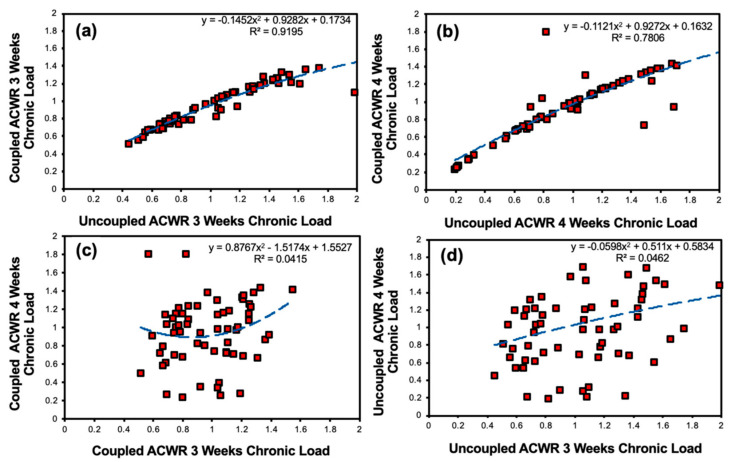
Relationships between coupled and uncoupled workloads with 3- or 4-week chronic workloads. (**a**) Coupled versus uncoupled when the chronic workload was measured over a 3-week period. (**b**) Coupled versus uncoupled when the chronic workload was measured over a 4-week period. (**c**) Coupled ACWR with a 4-week chronic workload versus chronic ACWR with a 3-week chronic workload. (**d**) Uncoupled ACWR with a 4-week chronic workload versus uncoupled ACWR with a 3-week chronic workload. The red squares represent the data and blue dashes displays the logarithmic best fit curve.

**Figure 12 sensors-24-01463-f012:**
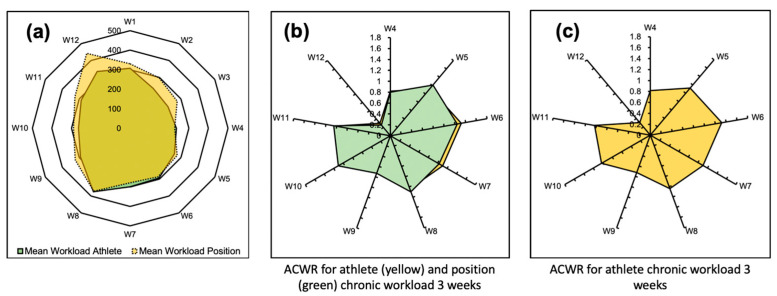
Dashboard in the form of spider charts to enable the comparison of the performance of a specific athlete with that of the overall team. The selection of the athlete was randomized by utilizing a randomization algorithm. The data compiled herein were computed via the Rolling Average model. (**a**) Comparison of the mean workload over the duration of the season for a specific athlete (green) versus the mean workload of the entire team (yellow). (**b**) Comparison of the ACWR over the duration of the season for a specific athlete (yellow) versus the mean ACWR of the entire team (green). (**c**) Comparative profile of the ACWR for the athlete for each week.

**Figure 13 sensors-24-01463-f013:**
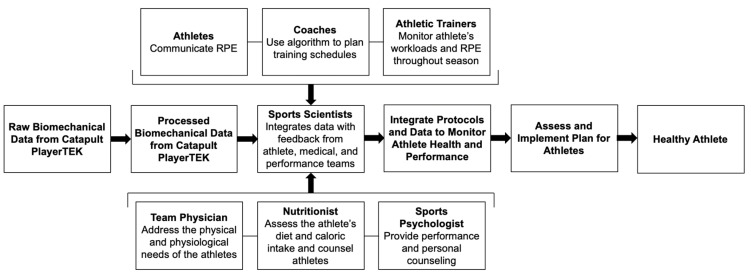
Operational process flow detailing the application and integration of data acquired from wearable technology as a complementary digital diagnostic for monitoring and informing athlete health and performance.

**Table 1 sensors-24-01463-t001:** Examples of wearable GPS technology sensor companies and unit costs.

Company	Device Name	Product Functionality	Sports	Unit Cost	Total Cost *
Catapult	Catapult Vector S7	Live tracking, heart rate, distance, inertial movements, sport specific analysis	Soccer, football, basketball, lacrosse	$1500	$37,500
	PlayerTEK	Live tracking, heart rate, distance, work ratio, acceleration/deceleration count		$219	$5475
	Catapult ONE	Distance, work-recovery ratio, impacts, metabolic power		$155	$3875
Fieldwiz	GPS FieldWiz V2 18Hz Fc	Distance, heart rate, acceleration and deceleration	Soccer, rugby	~$483.20	~$12,079.40
	GPS FieldWiz V2 18Hz	Distance, acceleration and deceleration		~340.70	~$8517.80
Gpexe	Gpexe pro2	Live tracking, distance, inertial movements	Soccer, rugby	$1260	$31,500
	Gpexe lt	Distance (position and velocity)		$630.75	$15,678.75
Inmotio	Inmotio GPS	Distance, velocity, acceleration and deceleration	Soccer, Football	N/A	
McLloyd	STv4 GPS Offline Data	Distance, velocity, acceleration, impact, biomechanics	Soccer, football	$40/month	$1000
	STv4 GPS + HR Offline Data	Distance, heart rate, velocity, acceleration, impact, biomechanics,		$50/month	$1250
	STv4 GPS Live Data	Live tracking, distance, velocity, acceleration, impact, biomechanics		$60/month	$1500
	STv4 GPS + HR Live Data	Live tracking, heart rate, distance, velocity, acceleration, impact, biomechanics		$70/month	$1750
Polar	Polar Team Pro	Distance, heart rate, top velocity, number of sprints, speed zones	Soccer, football	N/A	N/A
Soccerbee	BEE	Distance, top velocity, number of sprints, game replay	Soccer	$168.99	$4224.75
	BEE Lite	Distance, top velocity, number of sprints		$128.99	$3224.75
SPT	SPT2 Pack	Heart Rate, distance, work rate, intensity	Soccer, football, lacrosse, rugby	$260	$6500
Stat Sport	Apex Athlete Series	Distance, maximum velocity, intensity and strain levels	Soccer, football, rugby	$299.99	$74,999.75
	Apex Team Series	Live tracking, distance, maximum speeds, intensity and strain levels, target thresholds		N/A	
Titan Sport	Titan 1+	Distance	Soccer, football	$150	$3750
	Titan 2	Live tracking of distance and inertial movements		$250	$6250
	Titan 2+	Live tracking of distance and inertial movements, 25 Hz sampling rate		$650	$16,250
Vx Sports	Vx Log	Live tracking, distance, work-recovery ratio, body force detection	Soccer, basketball	$349	$8725
WIMU	WIMU Pro Elite Tracking System	Distance, high metabolic load distance, maximum velocity, number of sprints, accelerations and deceleration, number of impacts	Soccer, rugby	N/A	N/A
Zebra	MotionWorks	Uses RFID; distance, velocity, orientation, acceleration	Football	N/A	N/A

* Extrapolation of the total cost assumes 25 athletes per team.

## Data Availability

The data may be made available upon reasonable request by contacting the corresponding author.

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
