# Peer review of "Wearable Devices and Digital Biomarkers for Optimizing Training Tolerances and Athlete Performance: A Case Study of a National Collegiate Athletic Association Division III Soccer Team over a One-Year Period"

_sensors, 2024, doi:10.3390/s24051463_

Round 1

Reviewer 1 Report

Comments and Suggestions for Authors

Very good work, well written with a good design and data analysis.

Author Response

We thank the reviewer for taking the time to go through our manuscript. We hope our manuscript is accepted for publication in this esteemed journal.

Reviewer 2 Report

Comments and Suggestions for Authors

The authors gathered normative baseline data on healthy male collegiate athletes at the DIII level toward development of future performance optimization protocols. I have the following problems:

1. The authors did not justify the relationship between the heat map color and the workload value.

2. What’s the H0 hypothesis of the t-value in the figure 4(a), 5(a), 6(a), 8(a), and 9(a)? And I could not understand the meaning of the figure 4(b), 5(b), 6(b), 8(b), and 9(b). What is the relationship between this plot with normality assessment? If the H0 hypothesis is that the distribution of workload of position group in Week N is same to that of the team workload (306.11±55.6), I suggest that the authors could adopt p-value instead of t-value. If the (b) was corresponding to the t-value, I suggested that the authors adopted 95% CI of the workload rather than a plot without value. 

3. In Figure 11, the authors should give the p-value of fit test.

4. The authors did not introduce the Table I and Figure 13 in the main text.

Author Response

We thank the reviewer for taking the time to go through our manuscript. We hope our manuscript is accepted for publication in this esteemed journal.

1) The objective of the heat map is for efficient viewing rather than the quantitative metric. Grouping in percentiles is meant to simplify. For specific numbers, the reader can refer to panels A and B. 

A qualitative color-gradient using green for low workload (10th percentile), yellow for medium workload (50th percentile), and red for high workload (90th percentile) permits for efficient viewing of the differences in weekly workload during the study (Fig. 1c).

2) Figures 4(b), 5(b), 6(b), 8(b) and 9(b) serve as a qualitative illustration for efficient viewing of the normality assessment values depicted in figures 4(a), 5(a), 6(a), 8(a), and 9(a). The null hypothesis is 0, indicating that there is no difference between the team's mean workload and the mean of each position group’s (e.g. attacking midfield, center back, forward) workloads each week. We used the t-value since it is a measure of the size of the difference relative to the variation in your sample data. Therefore, these values are the calculated difference represented in units of standard error. The greater the magnitude of T, the greater the evidence against the null hypothesis. We have included t-values since we are looking at ratios rather than a p-value. We are not trying to determine statistical significance, but rather qualitatively show differences in workloads each week during the season. 

The null hypothesis was 0, indicating no difference between the team’s mean workload and the workload for the respective position group’s workload. A greater positive magnitude suggests a higher average workload compared to the mean, whereas a greater magnitude of the negative t-values indicates a negative deviation from the mean workload, thus a lower workload.

3) We have now included p-values into the results section:

A relationship between the coupled versus uncoupled workload when the chronic workload was measured over a 3-week period was determined (R2 = 0.92, P = 0.10) (Fig. 11a). The relationship between the uncoupled ACWR and coupled ACWR across a 4-week period was found (R2 = 0.78, P = 0.17) (Fig. 11b). A weak relationship between the coupled ACWR across a 3-week chronic workload and the coupled ACWR across a 4-week chronic workload was additionally determined (R2 = 0.04, P = 0.41) (Fig. 11c). The relationship between the uncoupled ACWR across a 3-week chronic workload and the uncoupled ACWR across a 4-week chronic workload was examined (R2 = 0.05, P = 0.39) (Fig. 11d).

4) Thank you for the suggestion. We have now included introductions to Table 1 and Figure 13.

Table 1 provides comparative analysis of the cost of current wearable GPS sensors used in sports today. 

The implications of this study further suggest the potential to create an athlete management system for coaches, athletic trainers, team physicians, and athletes to follow in future seasons (Fig. 13).

Reviewer 3 Report

Comments and Suggestions for Authors

Dear Authors,

In my opinion, the article is well written, but I suggest the following improvements:

1. In the summary, better explain the objective of the work.

2. I suggest adding a second paragraph associated with the Methodology of the work.

3. They do not include the final section Conclusions and future work. Indicate the main conclusions in a qualitative way and later in a quantitative way according to the results obtained.

4. It is suggested to increase the number of references from the last two years (2022 and 2023).

Author Response

We thank the reviewer for taking the time to go through our manuscript. We hope our manuscript is accepted for publication in this esteemed journal.

1. We have reworded the objective of the work:

The objective of this retrospective study was to gather big data using Catapult wearable technology, develop an algorithm for musculoskeletal modeling, and longitudinally determine workloads of male college soccer (football) athletes at the Division III (DIII) level over the course of a 12-week season

2. We thank the reviewer for the feedback regarding the Methodology of the work, we have implemented the suggested comments. See manuscript for updated revision.

3. We thank the reviewer for their suggestions to better organize the sections and have responded to the feedback, including a Conclusions and Future Work section. See manuscript for updated revision.

4. We thank the reviewer for the suggestion. We have added 10 papers from more recent years. Please see references for updated revision.

Reviewer 4 Report

Comments and Suggestions for Authors

The research paper provides valuable insights into optimizing athlete health management through wearable sensors. The authors highlight the current lack of research on workload and potential injury risk, and they draw conclusions based on data analysis. The utilization of wearable sensors for athlete load analysis as a supplementary diagnosis to monitor athlete health and performance presents a novel, forward-looking approach to sports management. While the manuscript is very well written, it should be modified before publication. I encourage that the authors address the following comments listed below:

1. Are there any potential challenges during the experimental process of collecting sensor data? If so, please discuss.

2. Please elaborate on the significance of key parameters mentioned in the paper, such as coupled and uncoupled ACWR.

3. In the Mateial and Method Part, more details on the methodology should be given.

4. In the abstract part, it seems the motivation does not well be states and the issues to address is missing.

5. Quality of several figures should be improved. For example, the labels are too small to read.

6. As a case study, could this findings be applicable to other similar studies in this field? 

Author Response

We thank the reviewer for taking the time to go through our manuscript. We hope our manuscript is accepted for publication in this esteemed journal.

1. Potential challenges during the experimental process are out of scope since this is a retrospective study. The data was generously given to us by our collaborators. 

2. Our motivation to use ACWR was based on prior clinical evidence: athletes who had an ACWR greater than 1.6 were 1.5 times more likely to suffer either a myotendinous or ligamentous injury.

The relationship between the uncoupled and coupled ACWR using the 3-week (21 day) chronic workload, a strong, positive relationship, presents findings consistent with previous exploration of this relationship[12]. Contextualization of workload with ACWRs for each position have implications for understanding the propensity for injury risk in the athletes of this study

3. We made changes to the Material and Methods part based on suggestions from another reviewer. 

4. Not much research has been conducted at lower collegiate division levels. Therefore, the current retrospective study aimed to gather big data, develop an algorithm for musculoskeletal modeling, and determine workloads of male college soccer (football) athletes at the Division III (DIII) level. We made changes to clarify the objective of the study part based on suggestions from another reviewer. 

5. Quality of several figures have been edited and improved. We thank the reviewer for the suggestion.

6. We have active projects including women’s soccer and this can be used for any rehabilitation application. Professional American Football has utilized this technology to calculate workload and examine soft tissue injury rates in order to monitor health and safety in professional athletes [1].  Collegiate athletics are increasingly using wearable technology to monitor workload as well, including a NCAA DI Women’s Soccer Team that looked at the external workloads calculated by GPS wearable devices and athlete’s subjective rated perceived exertion score [2]. This type of study has also been done globally in English rugby and Australian football [3,4]. Our case study applies a similar methodology to lower collegiate division athletes. Literature regarding monitoring workload in this sample is limited, thus our study’s objective is to highlight this population. 

[1] R. T. Li, M. J. Salata, S. Rambhia, J. Sheehan, and J. E. Voos, “Does Overexertion Correlate With Increased Injury? The Relationship Between Player Workload and Soft Tissue Injury in Professional American Football Players Using Wearable Technology,” Sports Health, vol. 12, no. 1, pp. 66–73, Aug. 2019, doi: 10.1177/1941738119868477.

[2] A. T. Askow et al., “Session Rating of Perceived Exertion (sRPE) Load and Training Impulse Are Strongly Correlated to GPS-Derived Measures of External Load in NCAA Division I Women’s Soccer Athletes,” J. Funct. Morphol. Kinesiol., vol. 6, no. 4, Art. no. 4, Dec. 2021, doi: 10.3390/jfmk6040090.

[3] Cousins BEW, Morris JG, Sunderland C, Bennett AM, Shahtahmassebi G, Cooper SB. Match and Training Load Exposure and Time-Loss Incidence in Elite Rugby Union Players. Front Physiol. 2019 Nov 19;10:1413. doi: 10.3389/fphys.2019.01413. PMID: 31803067; PMCID: PMC6877544.

[4] Boyd LJ, Ball K, Aughey RJ. Quantifying external load in Australian football matches and training using accelerometers. Int J Sports Physiol Perform. 2013 Jan;8(1):44-51. doi: 10.1123/ijspp.8.1.44. Epub 2012 Jul 31. PMID: 22869637.

Round 2

Reviewer 4 Report

Comments and Suggestions for Authors

The authors addressed all the previous comments. This revised manuscript is ready for publication.

Comments on the Quality of English Language

Excellent